# Molecular Characterization, Expression Pattern, DNA Methylation and Gene Disruption of *Figla* in Blotched Snakehead (*Channa maculata*)

**DOI:** 10.3390/ani14030491

**Published:** 2024-02-01

**Authors:** Yang Zhang, Yuntao Lu, Feng Xu, Xiaotian Zhang, Yuxia Wu, Jian Zhao, Qing Luo, Haiyang Liu, Kunci Chen, Shuzhan Fei, Xiaojuan Cui, Yuandong Sun, Mi Ou

**Affiliations:** 1School of Life Sciences, Hunan University of Science and Technology, Xiangtan 411201, China; dy211004@163.com (Y.Z.); 17633537502@163.com (Y.L.); xjcui@hnust.edu.cn (X.C.); 2Key Laboratory of Tropical and Subtropical Fishery Resources Application and Cultivation, Ministry of Agriculture and Rural Affairs, Pearl River Fisheries Research Institute, Chinese Academy of Fishery Sciences, Guangzhou 510380, China; zxt13733320610@163.com (X.Z.); wyx07260726@163.com (Y.W.); zhaojian@prfri.ac.cn (J.Z.); luoqing@prfri.ac.cn (Q.L.); hyliu@prfri.ac.cn (H.L.); chenkunci@prfri.ac.cn (K.C.); feisz@prfri.ac.cn (S.F.); 3Chongqing Fisheries Technical Extension Center, Chongqing 404100, China; xufenghubei@163.com

**Keywords:** *Figla*, ovarian development, gene expression, epigenetic regulation, CRISPR/Cas9 system

## Abstract

**Simple Summary:**

There is a significant sexual dimorphism in the blotched snakehead (*Channa maculata*). However, studies on sex differentiation and gonadal development of *C. maculata* are limited. Factor in the germ line alpha (Figla) is an oocyte-specific basic helix–loop–helix (bHLH) transcription factor, involved in ovarian development. In the present research, the expression patterns and cellular location of *Figla*, as well as the CpG methylation of its promoter, demonstrated that Figla participated in ovarian differentiation and development in *C. maculata*. Meanwhile, the gene disruption of *Figla* was realized by the CRISPR/Cas9 system for further functional research.

**Abstract:**

*Figla* is one of the earliest expressed genes in the oocyte during ovarian development. In this study, *Figla* was characterized in *C*. *maculata*, one of the main aquaculture species in China, and designated as *CmFigla*. The length of *CmFigla* cDNA was 1303 bp, encoding 197 amino acids that contained a conserved bHLH domain. *CmFigla* revealed a female-biased expression patterns in the gonads of adult fish, and *CmFigla* expression was far higher in ovaries than that in testes at all gonadal development stages, especially at 60~180 days post-fertilization (dpf). Furthermore, a noteworthy inverse relationship was observed between *CmFigla* expression and the methylation of its promoter in the adult gonads. Gonads at 90 dpf were used for in situ hybridization (ISH), and *CmFigla* transcripts were mainly concentrated in oogonia and the primary oocytes in ovaries, but undetectable in the testes. These results indicated that Figla would play vital roles in the ovarian development in *C. maculata*. Additionally, the frame-shift mutations of *CmFigla* were successfully constructed through the CRISPR/Cas9 system, which established a positive foundation for further investigation on the role of *Figla* in the ovarian development of *C. maculata*. Our study provides valuable clues for exploring the regulatory mechanism of *Figla* in the fish ovarian development and maintenance, which would be useful for the sex control and reproduction of fish in aquaculture.

## 1. Introduction

The bHLH superfamily contains numerous transcription factors that are essential for gene regulation in most eukaryotes [1], such as sex determination [2], cell proliferation, and differentiation [3]. Figla is an oocyte-specific bHLH transcription factor, which has been reported to play crucial roles in oocyte growth and development [4]. In mice (*Mus musculus*), *Figla* transcripts can be first detected at embryonic stage 13.5, which is restricted to oocytes [5]. *Figla* can promote the formation of primordial follicles and activate the expression of oocyte-related genes. After knockout of *Figla* in *M. musculus*, oocyte apoptosis was observed and all primordial follicles were lost right after birth [5], and genes related to germ cell differentiation and oogenesis were significantly reduced, such as *Lhx8*, *Sohlh1*, and *Gdf9* [6]. Additionally, *Figla* was thought to be one of the genes that caused premature ovarian failure in women [7].

*Figla* was also identified in several teleosts, and displayed the sexually dimorphic expression pattern in the gonads, which was dominantly expressed in the ovaries. In Nile tilapia (*Oreochromis niloticus*), *Figla* mRNA was remarkably expressed during ovarian development, especially in the early primary oocytes, while no specific signals could be detected in testes [8]. In Japanese flounder (*Paralichthys olivaceus*), *Figla* showed a higher expression in ovaries than in testes, and *Figla* was specifically expressed in germ cells, in which *Figla* signals were largely detected in oocytes and weakly in spermatogonia and primary spermatocytes [9]. *Figla* expression was also changed during sex transformation. In the protogynous wrasse (*Halichoeres poecilopterus*), *Figla* expression levels dwindled during the female-to-male sex change, accompanied by oocytes degeneration [10]. In protandrous black porgy (*Acanthopagrus schlegelii*), *Figla* was even presumed to induce the transformation of Sertoli cells into follicle-like cells during the male-to-female sex change [11]. Similar to the study in *M. musculus*, the gonads were masculinized after the disruption of *Figla* in zebrafish (*Danio rerio*), resulting in an all-male phenotype in the mutants [12]. Furthermore, the over-expression of *Figla* in XY *O. niloticus* led to defective spermatogenesis, accompanied by the attrition of meiotic spermatocytes as well as spermatids [8]. These results indicated that *Figla* also played a key role in the ovarian differentiation and development of fish.

Epigenetic regulation is thought to play an important role in gonadal differentiation and development, in which DNA methylation has been considered as a vital actor [13]. Studies have indicated that the relationship between *Figla* expression and the methylation of its promoter plays a vital role in gonadal development. In half-smooth tongue sole (*Cynoglossus semilaevis*), *Figla* was less expressed and its promoter was highly methylated in male individuals, whereas in the pseudomales and females, the expression levels of *Figla* were significantly higher than that in males, and the methylation status was lower than that in males [13]. In channel catfish (*Ictalurus punctatus*), the expression of *Figla* was low, with high methylation levels in males; however, it became progressively demethylated after 17β-oestradiol exposure. Meanwhile, *Figla* expression was also gradually increased [14]. Therefore, it is necessary to investigate whether DNA methylation affects gonadal development by regulating the expression of *Figla* in fish.

Blotched snakehead (*Channa maculata*) is well received by consumers for its delicious flavor, high protein content, few intramuscular spines, and medicinal properties. Males grow much faster than females; therefore, an all-male population is greatly beneficial for aquaculture. In our previous study, XY sex-reversal females (XY-F) and YY super-males (YY-M) were produced by the combination of 50 mg/kg 17β-estradiol induction and a sex-specific molecular marker [15]. However, the mechanisms of sex differentiation and gonadal development in *C. maculata* was unclear. The gonadal histology and transcriptome analysis of 6-month-old *C. maculata* indicated that the ovarian development in *C. maculata* was asynchronous, where oocytes of all stages of development were present. In addition, *Figla* was shown to be a female-biased gene in *C. maculata*, whose expression was dominantly higher in the ovaries of normal XX females (XX-F) and XY-F than in the testes of normal XY males (XY-M) and YY-M [16]. Therefore, we would like to know if Figla also plays a role in the ovarian development in *C. maculata*, similar to studies in other fish. In this study, the *Figla* gene from *C. maculata* was cloned and analyzed, and the expression patterns of *Figla* were expounded in different adult tissues and gonads at different developmental stages. Furthermore, the CpG methylation levels of *Figla* promoter were also illuminated in gonads. In order to further deepen the functional research of *Figla* in the future, the CRISPR/Cas9 system was used to construct *Figla* mosaic F0 generation. This study explores the role of *Figla* in the gonadal differentiation and development, and is expected to provide a reference for the study of the sex differentiation and gonadal development mechanism of blotched snakehead.

## 2. Materials and Methods

### 2.1. Experimental Fish and Sampling

Experimental fish were reared in the fish laboratory of the Model Animal Research Center, Pearl River Fisheries Research Institute (Guangzhou City, Guangdong Province, China). Adult XY-M, XX-F, XY-F, and YY-M individuals (*n* = 3) were determined by sex-specific molecular marker and gonadal histology [15]. After anesthesia with MS222 anesthetic (Sigma-Aldrich, St. Louis, MO, USA), at a dose of 1.25 g/mL, the multitudinous tissues, such as liver (L), gills (G), middle kidney (MK), spleen (S), head kidney (HK), intestines (I), muscle (M), heart (H), gonads (ovaries or testes (O/T)), brain (B), hypothalamus (Hy), and pituitary (P) were collected from one-year-old XX-F, XY-F, XY-M, and YY-M individuals (*n* = 3), respectively. These tissue samples were used for gene cloning, expression, and DNA methylation analysis.

The gonads from XX-F and XY-M individuals (*n* = 3) at different developmental stages (30, 60, 90, 120, 150, 180, and 365 dpf) were obtained for measuring the expression levels of *Figla* during the developmental stages of gonadal differentiation, and the genetic sex was identified as mentioned above. All samples were quickly frozen in liquid nitrogen and stored at −80 °C. Additionally, the gonads from 90 dpf XX-F and XY-M individuals were fixed in Bouin’s solution (ABI, Foster City, CA, USA) for ISH. All fish operations complied with the regulations of the National Institutes of Health guide for the care and use of laboratory animals.

### 2.2. RNA Extraction and cDNA Synthesis

Total RNAs of the above samples were isolated by TRIzol (Invitrogen, Carlsbad, CA, USA). The RNA quality and quantity were determined by spectrophotometer (Thermo Fisher, Waltham, MA, USA). Then, 1 μg RNA was electrophoresed with a 1.0% agarose gel to analyze the RNA integrity. The remaining RNA sample was frozen at −80 °C. The 5′-RACE-Ready and 3′-RACE-Ready cDNA were obtained by SMART^™^ RACE cDNA Amplification Kit (TaKaRa, Otsu, Japan). cDNA was synthesized by the ReverTra Ace qPCR RT Kit (Toyobo, Osaka, Japan) [17]. All operations were performed according to the manufacturer’s instructions.

### 2.3. Full-Length cDNA Cloning and Sequence Analysis

The predicted cDNA sequence of *Figla* was obtained from the genome of blotched snakehead (SRA Accession No. PRJNA730430). Figla-F1/R1 primer pair was designed by Primer 5.0 to identify the predicted cDNA sequence of *Figla*. Then, 5′-RACE and 3′-RACE primers were designed for PCR amplification with 5′-RACE-Ready and 3′-RACE-Ready cDNA as templates to amplify 5′ and 3′untranslated regions (UTRs), respectively. Thereafter, the full-length cDNA sequence of *Figla* was amplified by Figla-F2/R2 primer pair with 5′ and 3′UTRs. The primers are showed in Table 1. PCR product containing the target fragment was purified by Gel Extraction Kit (Omega Bio-Tek, Norcross, GA, USA), purified product was ligated into pMD19-T vectors (TaKaRa, Japan), and the ligated product was transformed into *Escherichia coli* DH5α competent cells (TransGen Biotech, Beijing, China). Positive colonies were selected and sequenced by a commercial company (Tianyi huiyuan, Guangzhou, China) [17].

Homologous Figla protein sequences from other species were gained from NCBI (https://www.ncbi.nlm.nih.gov/, accessed on 23 October 2023), including *C. argus*, *C. striata*, *Lates calcarifer*, *L. japonicus*, *C. semilaevis*, *Oryzias latipes*, *D. rerio*, *Xenopus tropicalis*, *Microcaecilia unicolor*, *Pelodiscus sinensis*, *Gopherus evgoodei*, *Gallus gallus*, *Strigops habroptila*, *Bos taurus*, *M. musculus*, and *Homo sapiens*. The deduced amino acid sequences of Figla in *C. maculata* and its homologous counterparts from other species were aligned using DNAMAN. Then, the neighbor-joining method was used to construct the phylogenetic tree of Figla proteins from different species using MEGA 11.0, and bootstrap values were calculated with 1000 replications.

### 2.4. Genome DNA Sequence Cloning and Structure Analysis

Genomic DNA was extracted from the tail using the Universal Genomic DNA Kit (CWBIO, Beijing, China). The predicted genome DNA sequence of *Figla* was obtained from the blotched snakehead genome (SRA Accession No. PRJNA730430). Specific primer pairs were designed to verify the predicted genome DNA sequence of *Figla* (Table 2). PCR products were recovered, ligated, transformed, and sequenced as mentioned above. The genomic structure of *Figla* was determined by comparing the genome DNA sequence with the cDNA sequence in conformity with the GT/AG principle. Softberry (http://www.softberry.com/berry.phtml, accessed on 24 October 2023) was applied to predict the transcription start site (TSS). Animal TFDB (http://bioinfo.life.hust.edu.cn/AnimalTFDB/, accessed on 24 October 2023) was used to predict the potential transcription factor binding sites (TFBS) [17].

### 2.5. Quantitative Real-Time PCR (qRT-PCR)

The primers for qRT-PCR are revealed in Table 1. The housekeeping genes *β-actin* and *EF1α* were utilized as internal controls for cDNA normalization [17]. qRT-PCR was implemented on the StepOnePlus™ Real-Time PCR System (ABI, USA) by SYBR^®^ Green Master Mix (Toyobo, Japan) according to the manufacturer´s instruction. The reaction system was (20 μL): 10 μL SYBR^®^ Green Master Mix, 0.8 μL for both forward and reverse primers, 7.4 μL ddH_2_O, and 1 μL cDNA. The qRT-PCR program was as follows: 95 °C for 2 min, 39 cycles of 95 °C for 15 s, 50 °C for 15 s, and 72 °C for 30 s. The default melting curve acquisition procedure of the instrument was used, 60 °C for 5 s and 95 °C for 5 s. qRT-PCR was performed for each sample in triplicates. Expression levels of *Figla* gene in different tissues and gonads at different developmental stage were calculated using the 2^−△△Ct^ method. For tissue expression, the expression level of the pituitary (P) in XX-F individuals was used as the baseline (1.0), and the expression levels of the different tissues in each type of fish were relative to that of the baseline. For gonad expression at different developmental stages, the expression level of the gonad in 30 dpf XX-F individuals was used as the baseline (1.0), and the expression levels of the gonads at different developmental stages were relative to that of the baseline.

### 2.6. In Situ Hybridisation (ISH)

The gonads of 90 dpf XX-F and XY-M were used for ISH analysis. Probe synthesis was performed as described by Feng et al. (2011) with minor modification [18]. The primers used for the amplification are displayed in Table 1. DIG RNA Labeling Mix (Roche, Mannheim, Germany) was employed for probe labeling, according to the manufacturer’s instructions. Additionally, the steps of ISH were conducted according to Gao et al. (2013) with slight emendation [19]. Briefly, gonad samples were dehydrated in a series gradient of ethanol and xylene before embedding in paraffin wax. The sections (7 μm thick) were fixed on amino slide and dried on a dryer (Kedee, Jinhua, Zhejiang, China) overnight at 40 °C. The next day, after the sections were dewaxed, each sample section was fixed with 4% paraformaldehyde (PFA) for 15 min and incubated with 10 μg/mL protease K at 37 °C for 10–15 min. The hybridization was carried out at 65 °C for 19 h (3 h for the pre-hybridization process and 16 h for the hybridization process). After hybridization, all sample sections were washed twice in mixed solution (5 mL 20 × SSC + 25 mL formamide deionized + 50 μL tween + distilled water, total 50 mL) at 65 °C for 30 min, 2 × SSC for 15 min, 0.2 × SSC washed twice for 30 min each time, and 1 × PBST three times for 5 min each time at room temperature. After samples were blocked in 2% blocking reagent for 3 h, they were incubated with blocking buffer mixed with anti-Digoxigenin-AP Fab fragments (Roche, Mannheim, Germany) for 16 h at 4 °C. Subsequently, all sample sections were washed six times in 1 × PBST for 10 min. After 5 min of NTMT equilibrium, they were then developed with NTMT mixed with DIG Nuleic Acid Delection Kit (Roche, Mannheim, Germany) for 3–5 h at room temperature. When the ideal chroma was reached, the sections were washed three times in 1 × PBST for 10 min each time to terminate the dyeing. The sections were covered with 50% glycerin, observed, and photographed using Nikon Eclipse Ti-U microscope (Nikon, Tokyo, Japan).

### 2.7. Bisulfite Sequencing PCR (BSP)

Gonads from the adult XX-F, XY-F, XY-M, and YY-M individuals (*n* = 3) were used for genomic DNA extraction and BSP analysis. BSP analysis was conducted as previously described [17]. In brief, genomic DNA from the gonads of each type of fish was extracted by the Universal Genomic DNA Kit (CWBIO, China) and mixed separately, and the EZ DNA Methylation-Gold^™^ Kit (Zymo Research, Irvine, CA, USA) was used to obtain the methylated DNA according to the manufacturer’s instructions. The DNA methylated CpG islands of *Figla* were predicted by MethPrimer 2.0 online software (http://www.urogene.org/cgi-bin/methprimer/methprimer.cgi, accessed on 25 October 2023), and the suitable CpG islands regions were selected as target fragments. Then, Figla-BSP-F/R primer pair was designed for PCR amplification (Table 1). The amplified DNA fragments were cloned into pMD19-T vectors and transformed into *E. coli* DH5α competent cells, and positive clones of each type of fish (*n* = 15) were randomly chosen and sequenced. CpG methylation percentage was calculated as (number of methylated sites)/(total number of CpG sites).

### 2.8. Knockout of Figla Gene in C. maculata

CRISPR/Cas9 system was used to knockout *Figla* in *C. maculata*. The target sites were designed using the CRISPRscan online tool (https://www.crisprscan.org/, accessed on 25 October 2023), and two guide RNAs (gRNAs) were designed to target *Figla* gene (Table 3). These gRNAs were synthesized using MAXIscript T7 In Vitro Transcription Kit (Invitrogen USA). Subsequently, gRNAs were purified by LiCl/ethanol precipitation method. gRNAs were severally mixed with TrueCut Cas9 Protein v2 (Invitrogen, USA) before microinjection, then 200 pg gRNA and 300 pg Cas9 protein was co-injected into 1–4 cell stage embryos of *C. maculata*. Five microinjected fry were collected after 48 h post-fertilization (hpf) and used for genomic DNA extraction. The primer pair Figla-JC-F/R was used for mutation detection (Table 1). The artificial insemination, microinjection, mutagenesis analysis, and fish rearing were based on our previous studies [20].

### 2.9. Statistical Analysis

The experimental data was exhibited as mean ± standard deviation (SD). All results were subjected to one-way ANOVA, followed by Dunnett’s test using SPSS Statistics 20.0. *p* < 0.05 was considered to be statistically significant. The correlation coefficient of CpG methylation percentage of *Figla* promoter and the mRNA expression of *Figla* was calculated using Excel 2021.

## 3. Results

### 3.1. Sequence Analysis of Figla in C. maculata

One *Figla* transcript was identified in *C. maculata*, named as *CmFigla*. The cDNA sequence of *CmFigla* was 1303 bp in long (GenBank accession no. PP067023), which included 431 bp 5′-UTR, 594 bp opening reading frame (ORF), and 278 bp 3′-UTR (Appendix A). *Cm*Figla encoded a putative protein of 197 amino acids (aa), and a highly conserved bHLH domain was predicted in 56-108 aa (Appendix A).

The multiple sequence alignments were conducted by DNAMAN, and the result revealed that Figla protein in vertebrates contained a conservative bHLH domain (Figure 1). *Cm*Figla had high similarities to Figla in Perciformes, which was 95.10% identical to *C. argus*, 90.86% to *L. japonicus*, 90.59% to *L. calcarifer*, and 85.99% to *C. striata*. Nevertheless, the similarities between *Cm*Figla and *Mm*Figla (Figla in *M. musculus*) and *Bt*Figla (Figla in *B. taurus*) were very low, only 23.08% and 11.56%, respectively. The phylogenetic tree showed Figla in *C. maculata* first clustered with Figla from the same genus species, *C. argus* and *C. striata*, then clustered with Figla from the same order, *L. calcarifer* and *L. japonicus*. Thereafter, they clustered with other teleosts, for example *O. latipes*, *C. semilaevis*, and *D. rerio*. Figla proteins from tetrapods clustered another clade, which was distant from the teleosts group, including *X. tropicalis*, *M. unicolor*, *P. sinensis*, *G. evgoodei*, *G. gallus*, *S. habroptila*, *B. taurus*, *M. musculus*, and *H. sapiens* (Figure 2).

By amplification, splicing, and alignment, the genome DNA sequence of *CmFigla* was verified with 8033 bp in long, which contained 929 bp 5′-flanking region, 6090 bp coding region and 1014 bp 3′-flanking region. The coding region covered five exons and four introns (Figure 3). A great deal of putative TFBS were predicted in *CmFigla*, including Estrogen Receptor 2 (Esr2), Sp1 Transcription Factor (Sp1), Sex-Determining Region Y Protein (SRY), and SRY-Box Transcription Factor 3 (SOX3), which were essential for the regulation of gene promoter activities. The CpG island of the *CmFigla* promoter was also predicted to be present (−445~+314) (Appendix A).

### 3.2. Expression Patterns of CmFigla

The expression levels of *CmFigla* in different tissues of XY-M, XX-F, XY-F, and YY-M individuals were detected by qRT-PCR. *CmFigla* transcripts were specifically expressed in gonads, especially in ovaries (O), whose expression was 215.33 ± 8.69-fold changes in the ovaries of XX-F, and 108.30 ± 4.21-fold changes in the ovaries of XY-F. In contrast, *CmFigla* mRNA was extremely low in the testes, with only 2.13 ± 0.14-fold changes in the testes of XY-M and 1.62 ± 0.12-fold changes in the testes of YY-M. Interestingly, there was also a small amount of *CmFigla* mRNA in the hypothalamus (Hy) (Figure 4a).

Figure 4b showed that the mRNA expression of *CmFigla* in the gonads existed an obvious sexual dimorphism, which was far higher in female ovaries than that in male testes at all developmental stages. In the ovaries, *CmFigla* transcripts sharply increased from 30 dpf (1.00 ± 0.17-fold changes), and peaked at 180 dpf (158.14 ± 11.43-fold changes), then decreased at 365 dpf (23.54 ± 1.18-fold changes). However, *CmFigla* always maintained very low expression levels at different testicular developmental stages, whose values ranged from 0.20 ± 0.03-fold changes (30 dpf) to 9.28 ± 0.03-fold changes (120 dpf).

### 3.3. CpG Methylation Levels of CmFigla Promoter

Differential CpG methylation levels of *CmFigla* promoter in the gonads from four types of fish were evaluated by BSP, and twenty-two CpG methylation sites were identified (Figure 5). The DNA methylation levels of *CmFigla* promoter were much lower in the ovaries of XX-F and XY-F individuals compared to those in the testes of XY-M and YY-M individuals. In the ovaries of XX-F and XY-F, the CpG methylation levels of *CmFigla* promoter were 20.6 ± 5.1% and 32.1 ± 6.5%, respectively. The CpG sites of *CmFigla* promoter maintained the hypermethylated status in the testes of XY-M and YY-M, with 86.1 ± 5.3% in XY-M and 89.7 ± 3.2% in YY-M. Meanwhile, most CpG sites of *CmFigla* promoter were not methylated at all in the ovaries of XX-F and XY-F. In XX-F ovaries, only locus 16 was fully methylated, while seven loci (11, 12, 14, 15, 17, 18, and 22) were completely unmethylated (Figure 5a). In XY-F ovaries, three loci (13, 15, and 16) had 100% methylation status, while eleven loci (1, 2, 5, 6, 7, 8, 9, 11, 12, 14, and 17) had 100% non-methylation status (Figure 5b). Conversely, the hypermethylation of *CmFigla* promoter resided in testes. Eight loci (3, 10, 12, 13, 14, 15, 16, and 21) in XY-M testes (Figure 5c) and fourteen loci (1, 2, 3, 4, 5, 11, 13, 14, 15, 16, 18, 19, and 21) in YY-F testes (Figure 5d) were wholly methylated, and no loci were not methylated.

Meanwhile, the correlation analysis indicated that the CpG methylation level of the *CmFigla* promoter was violently negatively correlated to *CmFigla* expression in the gonads of *C. maculata*, and the correlation coefficient (*R*^2^) was 0.9053 (Appendix A).

### 3.4. Cellular Location of CmFigla in Gonads

To confirm the cellular localization of the *CmFigla* transcripts, ISH analysis was performed in the gonads. In the ovaries of 90 dpf XX-F individuals, *CmFigla* signals were dominantly expressed in oocytes, and the intensity of the positive signals in oogonia and primary oocytes seemed to be stronger than that in growing oocytes (Figure 6a,b). There were no obvious signals observed in the germ cells of the XY-M testes (Figure 6d,e), in accordance with the results of qRT-PCR (Figure 4b). There were no signals to be detected in the negative controls with sense probe hybridization (Figure 6c).

### 3.5. Generation of the CmFigla Mutantion Using the CRISPR/Cas9 System

To induce *CmFigla* disruption, two CRISPR/Cas9 target sites were designed within the mature mRNA sequence of *CmFigla* located in Exon 2 and Exon 3 (Figure 3). The mutation analysis of the target sites in CRISPR/Cas9 system were performed by PCR amplification, sub-cloning, and sequencing in 48 hpf fry. By aligning mutated DNA sequences with wild-type sequences, it was found that gRNA1 located in Exon 2 did not cause mutation, but the gRNA2 located in Exon3 generated four forms of deletion: −58, −17, −6, and −5 bp (Figure 7a), and the mutagenesis frequency was 45.0% (9 mutated clones/20 total sequenced clones). Missing 6 bp and 58 bp resulted in the deletion of two and fifteen amino acids, respectively. Meanwhile, the other two forms of deletion (−5, −17 bp) led to the premature termination of transcription due to frame-shift mutations (Figure 7b). These results demonstrated that *Figla-gRNA* had the ability to edit the *Figla* gene in *C. maculata*, and mutation types and frequency indicated that most forms of deletion brought about frame shifts, leading to the disruption of gene structure.

## 4. Discussion

As an oocyte-specific transcription factor, Figla plays an essential role in the formation of primordial follicles [6], the transition from primordial to primary follicle [4], and gonadal development and differentiation [21]. In the present study, one homology Figla containing the conservative bHLH domain was illustrated in *C. maculata*, and named *CmFigla*. In most fish, such as *P. olivaceus* [22] or turbot (*Scophthalmus maximus*) [23], only one *Figla* transcript exists. However, two transcriptional variants (*Figla_tv1* and *Figla_tv2)* had been identified in *C. semilaevis* [24], and the difference between the two transcriptional variants was that Figla_tv2 lacked the bHLH region. In order to know whether multiple transcripts existed in other fish, *Figla* sequences in most teleosts were obtained from NCBI (https://www.ncbi.nlm.nih.gov/, accessed on 23 December 2023). It was found that there was more than one transcript in some fish, such as two forms in climbing bass (*Anabas testudineus*) (GenBank accession no. XM_026343723.1, XM_026343724.1), four forms in *I. punctatus* (GenBank accession no. XM_047149866.2, XM_017452432.3, XM_047149867.2, XM_047149868.2), three forms in Siamese fighting fish (*Betta splendens*) (GenBank accession no. XM_055511459.1, XM_029164147.3, XM_029164148.3), and so on. Similar to *C. semilaevis*, there were two different Figla protein in large yellow croaker (*Larimichthys crocea*) (GenBank accession no. XP_019121559.1, XP_027132394.1), river trout (*Salmo trutta*) (GenBank accession no. XP_029617814.1, XP_029617813.1), lake whitefish (*Coregonus clupeaformis*) (GenBank accession no. XP_041691711.1, XP_041691710.1), flier cichlid (*Archocentrus centrarchus*) (GenBank accession no. XP_030593618.1, XP_030593617.1, XP_030593616.1), sockeye salmon (*Oncorhynchus nerka*) (GenBank accession no. XP_029534804.1, XP_029534805.1), and Atlantic salmon (*Salmo salar*) (GenBank accession no. XP_014017212.1, XP_045560154.1); one Figla protein was common form, and the other lacked the conserved bHLH region. Perhaps different *Figla* transcripts were discovered due to the depth or coverage of sequencing, or the teleost-specific third round of whole-genome duplication [25]. Of course, more studies are needed to verify the possibility in the future. The protein sequence alignments indicated that the similarities of Figla between *C. maculata* and other fish species were high, and those between teleosts and tetrapods were low; similarly, the phylogenetic relationship of Figla proteins was in accordance with the evolutionary distance among the organisms.

*Figla* has shown an obvious sexual dimorphic expression pattern in the gonads of most teleosts. In *O. niloticus*, *Figla* expression was absolutely dominant in the early primary oocytes in ovaries, but almost undetectable in testes [8]. In *S. salar*, *Figla* showed significantly higher expression in the ovaries compared to the testes by qRT-PCR, and ISH indicated that *Figla* specific signals were only present within the oocytes [26]. Extensive transcriptome analysis of gonads also confirmed that *Figla* was female-biased, such as spotted scat (*Scatophagus argus*) [27] and tiger grouper (*Epinephelus fuscoguttatus*) [28]. Consistently, *CmFigla* expression in the ovaries was significantly higher than that in the testes, the intensity of the positive signals of *CmFigla* in oogonia and primary oocytes seemed to be stronger than that in growing oocytes, and negligible expression could be detected in the testes (Figure 4a and Figure 6). Furthermore, *CmFigla* expression in XY individuals could have been changed after 17β-estradiol induction; *CmFigla* showed high expression levels in the gonads of XY-F that was similar to that in the gonads of XX-F, which was different from the gonads of XY-M. Similar results have been observed in Japanese eel (*Anguilla japonica*) [29], *A. schlegelii* [11], and *I. punctatus* [14], in which *Figla* expression was up-regulated during ovarian differentiation after estrogen induction. These results revealed the conservative role of *Figla* in the ovarian development of *C. maculata*. Two transcripts (*Figla_tv1* and *Figla_tv2*) were identified in *C. semilaevis*; *Figla_tv1* was mainly expressed in the ovaries, skin and liver, but *Figla_tv2* was detected in all examined tissues, especially in male muscle [24]. In addition, there was considerable expression in the female brain of *P. olivaceus* [22], indicating that *Figla* may also play a role in other tissues. In our study, *CmFigla* was also detected in the male hypothalamus (Hy), indicating a role for *CmFigla* in hormone production through the hypothalamic–pituitary–gonadal (HPG) axis, which requires more study.

Many studies illustrated that *Figla* was essential during ovarian differentiation and development. In *O. niloticus*, *Figla* expression was significantly up-regulated within 30 days after hatching (dah) ovaries, then peaked at 90 dah and descended from 120 dah [8]. In *C. semilaevis*, *Figla_tv1* mRNA was first examined at 120 dah in ovaries and reached the maximum at 1 year after hatching (yah) before sharply declining at 2 yah [24]. In *P. olivaceus*, *Figla* expression in ovaries ascended from 8-month-old stage to 1-year-old and 1.5-year-old stage [9]. In sablefish (*Anoplopoma fimbria*), *Figla* expression was significantly increased when XX females reach 120 mm, which coincided with the emergence of CN oocytes, and with the development of PN oocytes [30]. Similarly, the expression of *CmFigla* could be detected at 30 dpf ovaries (the onset of ovarian differentiation), then gradually raised from 60 dpf and reached to the peak at 180 dpf (the critical stage of ovarian development). Subsequently, *CmFigla* expression descended to a similar level of 60 dpf at 365 dpf (ovulation stage) (Figure 4b). These finding indicated that *Figla* is closely related to differentiation and development of ovary in *C. maculata.*

DNA methylation has been considered as a vital actor in sex determination and gonadal development, which also regulates the expression of numerous genes [31]. The expression of *Figla* was low in male testes of *I. punctatus* with high methylation levels of *Figla* promoter; however, it became progressively demethylated after E2 exposure [14]. In *C. semilaevis,* the expression levels of *Figla* were significantly higher in pseudomales and females than that in males, and the methylation status of *Figla* promoter was lower in pseudomales and females than that in males [13]. Analogous findings were observed in our study, wherein it was found that the CpG methylation levels of *CmFigla* promoter in XY-M and YY-M testes were high, whereas XY-F and XX-F ovaries showed abundant *CmFigla* expression, and the CpG methylation levels of *CmFigla* promoter were significantly reduced compared to those in the testes (Figure 5). There was also a violently negative correlation between the CpG methylation level of *CmFigla* promoter and *CmFigla* expression in the gonads (Appendix A); similar results have been reported in *M. musculus* [32] and *D. rerio* [33]. It was suspected that DNA methylation might regulate the expression of *Figla* and participate in ovarian development and maintenance in *C. maculata*.

The deletion of the *Figla* gene in vertebrates usually leads to ovarian dysfunction. In *O.latipes*, two TALENs targets led to premature truncations and the bHLH region was destroyed, thus XX *Figla*^−/−^ mutants did not form follicles and the expression of female-specific genes (*Gdf9* and *Bmp15*) was reduced [34]. However, XX *Figlα*^−/−^ medaka obtained by CRISPR/Cas9 developed ovarian lamellae and ovarian cavities, but the *Figla* deficient oocytes were blocked at or around pachytene [35]. The destruction of *Figla* in *D. rerio* blocked the transition from cystic CN oocytes to individual follicular perinucleolar oocytes, resulting in an all-male phenotype in the homozygous mutant [12]. In this study, we introduced CRISPR/Cas9 system to construct *CmFigla* mutants. The designed gRNA effectively worked at the target sites, and four forms of deletion were obtained. The majority of deletions were frame-shift mutations that generated a premature termination codon, underwent early termination in translation, and disrupted the molecular functions of the *Cm*Figla protein. Successful knockout of *CmFigla* established a research basis for further investigating the role of *CmFigla* in the ovarian development of *C. maculata.* Due to the young age of the current gene knockout fish, two consecutive generations were required to obtain *CmFigla*^−/−^ homozygous mutants; thus, there were no results on whether the deletion of *Figla* will affect the gonadal development and oogenesis related genes expression in *C. maculata*, and we will continue to conduct the relevant studies in the future.

## 5. Conclusions

In conclusion, we characterized and analyzed the *CmFigla* gene in this study. *Cm*Figla has a conserved bHLH domain, and its sequence characteristics and gene expression patterns are similar to those reported in vertebrates. *CmFigla* revealed consumingly female-biased expression patterns in the gonads; furthermore, a noteworthy inverse relationship was observed between *CmFigla* expression and the CpG methylation status of its promoter in the gonads. *CmFigla* expression also indicated obvious sexual dimorphism at all gonadal development stages, which was far higher in female ovaries than in male testes. ISH proved again that *CmFigla* transcript was only detected in oocytes. According to these results, it was concluded that *Figla* plays a vital role in the ovarian differentiation and development in *C. maculata*. Additionally, we successfully constructed frame-shift *CmFigla* mutations through the CRISPR/Cas9 system, which established a positive foundation for further investigation on the role of *CmFigla* in the ovarian development of *C. maculata.* Our study provides valuable clues for exploring the potential function of *Figla* in fish ovarian development and maintenance, which may be useful for the sex control and reproduction of fish in aquaculture. Simultaneously, our study demonstrates that the CRISPR/Cas9 system is an effective tool for genetics studies in *C. maculata*, promoting genetic engineering breeding and gene function research in *C. maculata*.

## Figures and Tables

**Figure 1 animals-14-00491-f001:**
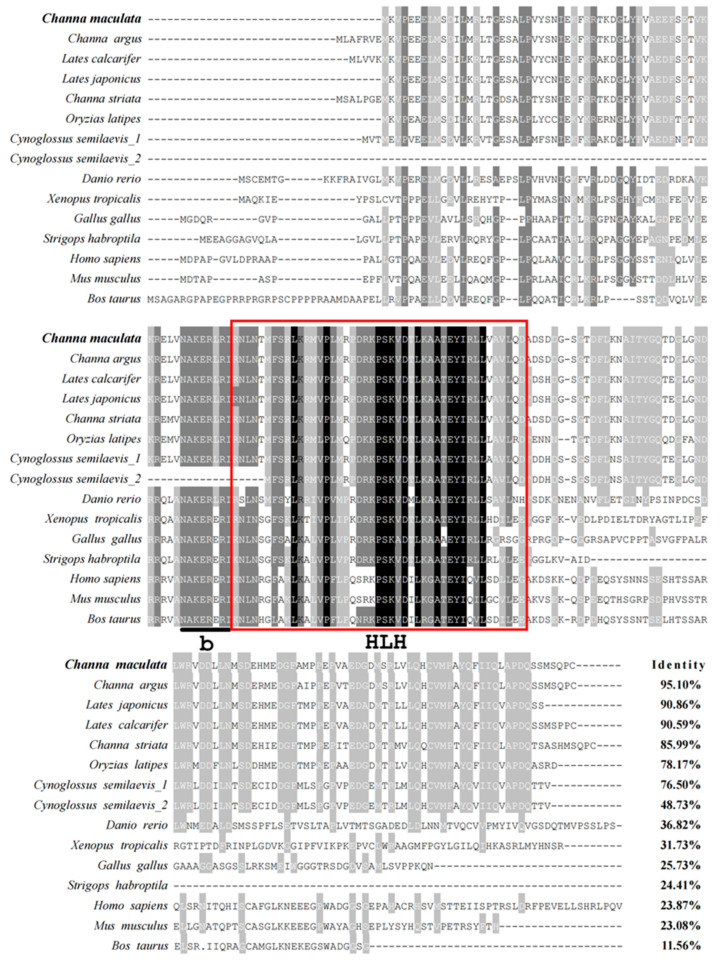
Multiple sequence alignments of Figla between *C. maculata* and other species. The presumed basic region (b) is underlined with black line and the conservative HLH domain is boxed in red.

**Figure 2 animals-14-00491-f002:**
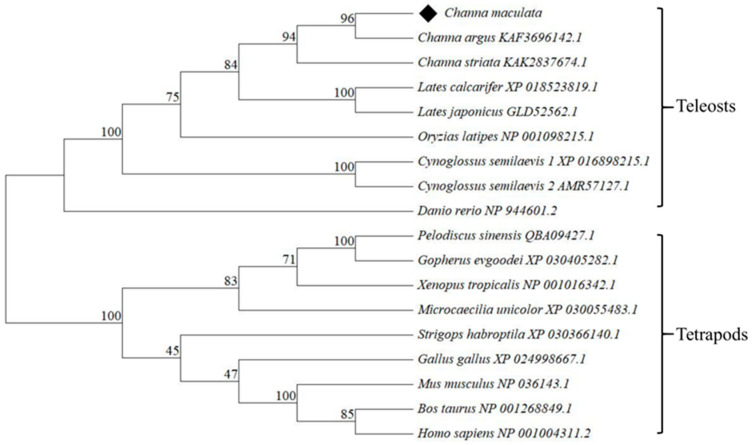
Phylogenetic analysis among the Figla amino acid sequences from different species. The bootstrap values were calculated with 1000 replications, and the bar indicates the distance. Figla of *C. maculata* was marked with a rhombus.

**Figure 3 animals-14-00491-f003:**
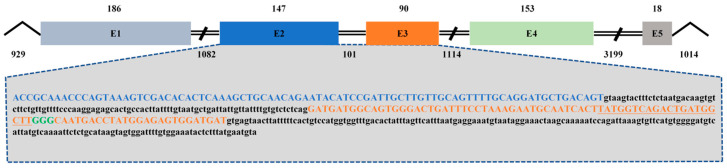
Genomic structure of *CmFigla* gene. Box indicates exons, and introns are between exons. The numbers above and below represent the lengths of exons and introns, respectively. CRISPR/Cas9 target sites are in Exon 2 and Exon 3, the gRNA target sites are shown as underscore followed by PAM in bold green.

**Figure 4 animals-14-00491-f004:**
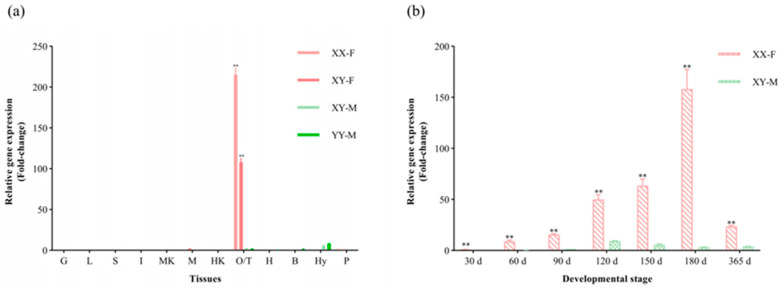
Gene expression analysis of *CmFigla* in different tissues of four types of *C. maculata* (*n* = 3) (**a**) and in the gonads of XX-F and XY-M individuals at different developmental stages (*n* = 3) (**b**). Values are expressed as mean ± SD. ** means extremely significant difference (*p* < 0.01).

**Figure 5 animals-14-00491-f005:**
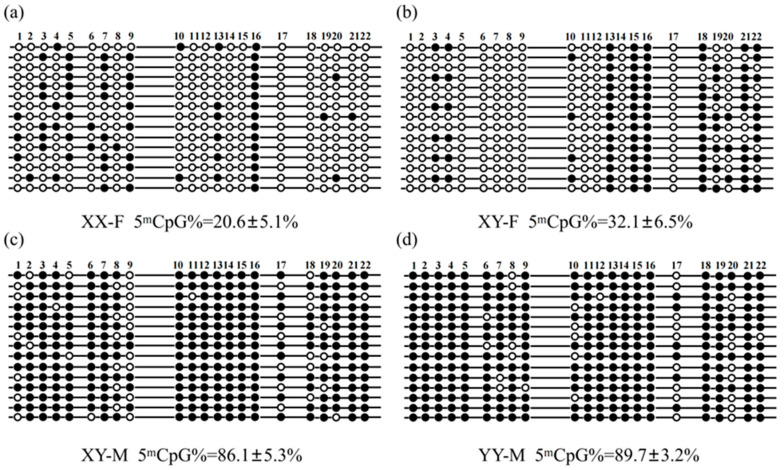
The CpG methylation profiles of *CmFigla* in gonads of adult XX-F (**a**), XY-F (**b**), XY-M (**c**), and YY-M (**d**) individuals (*n* = 3). Values are expressed as mean ± SD. The hollow circles mean the unmethylated CpG loci, and the solid circles mean the methylated CpG loci.

**Figure 6 animals-14-00491-f006:**
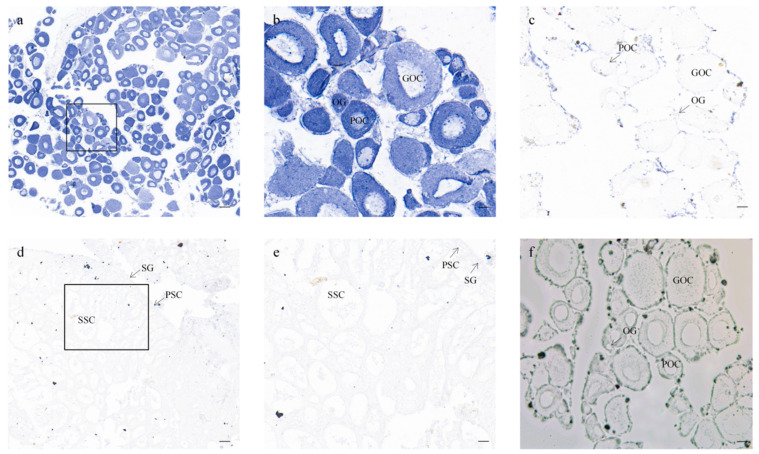
Cellular localization of *CmFigla* mRNA in gonads visualized by ISH. Sections of ovary (**a**–**c**,**f**), sections of testis (**d**,**e**). (**a**,**b**) and (**d**,**e**) were labeled with *CmFigla* antisense probe. Positive signals with the antisense probe were indicated in blue (**a**,**b**), panels (**b**) and (**e**) are the magnifications of black box in panels (**a**) and (**d**), respectively. The negative control with sense probe hybridization is (**c**). The arrows indicated the location of cell. The blank control is (**f**). OG: oogonia, POC: primary oocytes, GOC: growing oocytes, SG: spermatogonia, PSC: primary spermatocyte, SSC: secondary spermatocyte. Scale bar = 50 μm (**a**), scale bar = 10 μm (**b**,**c**,**e**,**f**); scale bar = 25 μm (**d**).

**Figure 7 animals-14-00491-f007:**
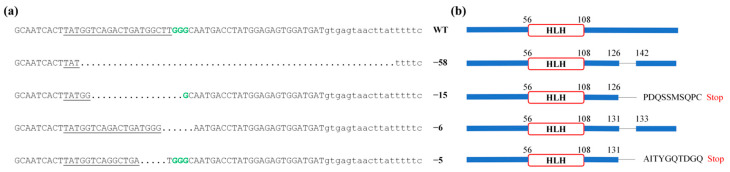
The schematic diagram of *CmFigla* in blotched snakehead. (**a**) CRISPR/Cas9 system-mediated modifications of *CmFigla* in *C. maculata.* The wild-type (WT) *CmFigla* gene sequences are shown on the top. The target sites of gRNA sequences are shown as underscore followed by PAM sequence (in bold green). Numbers in bracket indicate the number of nucleotides deleted (−) after editing. (**b**) Schematic diagrams show the predicted truncated proteins that would be produced from the mutated blotched snakehead. The numbers show the positions of amino acid residues. The loss of amino acid is displayed with single black lines in the domains.

**Table 1 animals-14-00491-t001:** Primers for full-length cDNA cloning, qRT-PCR, BSP analysis, and ISH.

Primer Name	Sequence (5′→3′)	Application
Figla-F1	CAGTCTGTCAGGTTGGGTAT	Partial sequence obtaining
Figla-R1	TTGGTACGCAGGCATCACA	
Figla-5′Rout	TACAAGCCGTCCTTAGTCCGTCTGAA	5′-Race PCR amplification
Figla-5′Rin	CCGTTAAACGCATCAAAATGTCACTC	
Figla-3′Fout	AGACTGATGGCTTGGGCAATGACCT	3′-Race PCR amplification
Figla-3′F in	CGGAGGATGGAGATATGAGCAGGC	
Figla-F2	GAGTGGAAAACACACTGC	ORF qualifying
Figla-R2	AGAGCAGATGGCTTCTAC	
Figla-BSP-F	TTTCGGTTTATTTCGTGAGTTT	DNA methylation analysis
Figla-BSP-R	ACCCTCTACCCACGACTACCT	
Figla-qF	CAACGCCAAGGAACGACTG	Quantitative real-time PCR
Figla-qR	CTCTCCATAAGTCATTGCCCA	
β-actin-qF	GCAAGCAGGAGTATGATGAG	
β-actin-qR	TTGGGATTGTTTCAGTCAGT	
EF1α-qF	GGGACACCCACAATAACATCC	
EF1α-qR	CCAGGCATACTTGAAGGAGC	
Figla-JC-F	ACCGCAAACCCAGTAAAGTC	Knockout detection
Figla-JC-R	TACATTCATAAAGAGTATTTCCACA	
Figla-ISH-F	CGGACTAAGGACGGCTTGT	In situ hybridization
Figla-ISH-R	AATTGGTACGCAGGCATCAC	
Figla-ISH-T7-zy-F	TAATACGACTCACTATAGGGCGGACTAAGGACGGCTTGT	
Figla-ISH-T7-zy-R	AATTGGTACGCAGGCATCAC	
Figla-ISH-T7-fy-F	CGGACTAAGGACGGCTTGT	
Figla-ISH-T7-fy-R	TAATACGACTCACTATAGGGAATTGGTACGCAGGCATCAC	

**Table 2 animals-14-00491-t002:** Primers for genome DNA sequences cloning.

Primer Name	Sequence (5′→3′)	Length (bp)
Figla-gDNA-F1	GAGTGTAAAGGGGTTGGTAC	1969
Figla-gDNA-R1	ATTCGCTTTTGAGATTTATGTA	
Figla-gDNA-F2	GTAGAAGTACCTCAAAAACCGT	1116
Figla-gDNA-R2	CACTTACGCTTGGCTTCTT	
Figla-gDNA-F3	GTCTAAGAACAGACTATGAGCTTGA	1841
Figla-gDNA-R3	TGTACGGCAAAGCCCAAT	
Figla-gDNA-F4	AGGATGTACCATGCCATAGTAG	1879
Figla-gDNA-R4	GATAGGGGTCCATGTAACAGTAC	
Figla-gDNA-F5	AACAAAGACGGCAGAGGTGAAT	1773
Figla-gDNA-R5	AACAGTCCATCTCCCTCCTCG	

**Table 3 animals-14-00491-t003:** The sequences of guide RNAs and the common primer used to target *CmFigla*.

Primer Name	Sequence (5′→3′)	Locus
Figla-gRNA1	TAATACGACTCACTATAGTGTCGACTTTACTGGGTTTGGTTTTAGAGCTAGAAATAGC	Exon 2
Figla-gRNA2	TAATACGACTCACTATAgTATGGTCAGACTGATGGCTTGTTTTAGAGCTAGAAATAGC	Exon 3
Common reverse	AAAAAAAGCACCGACTCGGT	/

Note: The underline sequences are the target sites.

## Data Availability

Data are contained within the article and Appendix A.

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
