# Peer review of "Molecular Characterization, Expression Pattern, DNA Methylation and Gene Disruption of Figla in Blotched Snakehead (Channa maculata)"

_animals, 2024, doi:10.3390/ani14030491_

Round 1
Reviewer 1 Report
Comments and Suggestions for Authors
PLS See attachment

Reviewer 2 Report
Comments and Suggestions for Authors
Line number:
19-37 Abstract
Please improve the writing so that it is clear and understandable. Ovary development or oogenesis are not always the same. Therefore, it is necessary from the abstract to make it clear exactly which phases oocytes were sampled and whether there is a separation between the different oocytes in oogenesis.
Any use of an abbreviation that appears for the first time should be written in detail and only then use the abbreviation. For example, here is the introduction and first appeared in the abstract. A basic helix–loop–helix (bHLH). Please correct in all the paper.
The abstract should be written so that the results of the work are as simple and clear as possible.
40 -104. Introduction
In the introduction, all the information appears, but in a confused way that makes it very difficult to understand the essence of the work.
I propose to emphasize in the introduction that it will be clear what type of oogenesis exists in Blotched snakehead (Channa maculata) and what are the hormones involved in the process of egg development which is the main subject of the study. The separation between the processes of sex determination and sex differentiation should be clear, maybe add a diagram because these processes in the studied fish are known.
The authors gave examples of similar processes in other fish but did not emphasize whether these processes are the same or different from Blotched snakehead.
Materials and methods
107-123. The writing should be improved so that it is clear, it was determined that the fish are normal XY male (XY-M), normal XX female (XX-F). Are there no studies describing and this is the first time here? Obviously, there is and suggests the authors bring citations of the process description.
133-144, 168 -181, 184-205, 207-220. Please add citations of works describing the process of MM or it is first time use this method.
Statistical processing should be in a separate chapter
Results
239-298. The writing of the results should be improved. I suggest that only the findings of this work appear in the results and all references should be forwarded for discussion. For example, transfer the drawings to the discussion: Figure 2 and 2.
I see that in the results there is a request for corrections.
Discussion
409-418. Results of studies on other species without citations.
439-453. Information to appear in the introduction please forward. In the discussion, the findings of this study should be compared to others and conclusions should be drawn, not detailed results.
